# Entering the Next Dimension: A Review of 3D User Interfaces for Virtual Reality

Adriel Yeo [1,2,†], Benjamin W. J. Kwok [1,2,†], Angelene Joshna [1], Kan Chen [1,2,*,†] and Jeannie S. A. Lee [1,2,*]

1   Infocomm Technology Cluster, Singapore Institute of Technology, Singapore 138683, Singapore;
    adriel.yeo@singaporetech.edu.sg (A.Y.); benjamin.kwok@singaporetech.edu.sg (B.W.J.K.);
    2002220@sit.singaporetech.edu.sg (A.J.)
2   Centre for Immersification, Singapore Institute of Technology, Singapore 138683, Singapore
*   Correspondence: kan.chen@singaporetech.edu.sg (K.C.); jeannie.lee@singaporetech.edu.sg (J.S.A.L.)
†   These authors contributed equally to this work.

**Abstract:** In the last decade, the use of virtual reality (VR) technologies has been increasing, as head-mounted display devices, such as the Meta Quest series, have become commercially accessible to everyday consumers. There are plenty of software applications developed for VR, ranging from games to serious training. Despite the fast-growing emergence of novel and natural gesture-based interface components, traditional windows, icons, menus, and pointer user interfaces still remain popular in these implementations. Therefore, navigating and interacting within 3-dimensional environments can still be challenging in virtual reality. This body of work will serve as a reference to the literature survey on 3-dimensional user interfaces and their implementations, and categorize them to reiterate a set of design recommendations for implementing 3-dimensional user interfaces in virtual reality. While questions revolve around the need for a 3-dimensional user interface if a 2-dimensional user interface suffices, we cannot ignore the capabilities and benefits offered when utilising 3-dimensional space. Unlike other work, this work proposes a novel metaphor-guided quadrant model to delineate the 3-dimensional user interface problem in an end-to-end manner. To achieve this, over 30 unique articles selected using a meta-analyses methodology were reviewed. The result highlights the dimensional semantics of the inputs and dimensional outcomes of the virtual targets. Notable reiterated design recommendations would match devices to interaction and then to the tasks in 3-dimensional environments, reducing the degrees of freedom, imposing constraints when possible, and not implementing one single best interaction technique. The results of this survey will be beneficial for virtual reality simulation designers or researchers attempting to determine the appropriate model when developing user interfaces for virtual reality content.

**Keywords:** virtual reality; 3D user interfaces; design guidelines

## 1. Introduction

Virtual reality (VR) has been explored for approximately the last fifty years, with the term *virtual reality* officially coined in the 1990s [1]. Since then, extensive research and studies have been conducted, paving the way for the modern head-mounted displays (HMDs) and VR headsets of today. For almost fifty years of research and development, VR technology has struggled to make significant breakthroughs [2]. It was not until the last decade that VR became accessible to everyday consumers with the release of affordable VR HMDs by companies such as Meta, Sony, Samsung, HTC, Apple, and Google, who are making big investments in VR and AR technology [3–5].

The accelerated interest in the platform and the *Metaverse* will contribute to a predicted significant boom in the VR market over the next few years, with possibly more commercial VR headsets that will enter the market, resulting in a surge of users and demand for VR applications [6]. However, there is no standardised aesthetic for 3-dimensional (3D) user

interfaces (UIs) compared to 2-dimensional (2D) graphical UI (GUI). There is therefore a need to consolidate 3D UI work such that the various techniques can be easily described, shared, and implemented by those interested in advancing the field [7]. The necessity for studying 3D UIs and reiterating UI design guidelines for VR is now higher than ever, to ensure an optimal user experience for consumers and to utilize the full potential of immersive technology. This also ensures minimum usability requirements for VR applications and promotes efficient and effective UIs. It also paves the way for cross-platform applications in the future, where applications can be compatible with different VR headsets or devices with minimal to no modifications.

Researchers across multiple fields [8–11] began creating applications for these devices, each with their own UI design. In these applications, users were required to interact with the hardware to provide input. They leverage pre-existing 2D design principles [12] such as traditional windows, icons, menus, and pointers (WIMP) interfaces even when designing for 3D interfaces. However, these 2D UI principles are sometimes not fully compatible with VR, as the 3D environment offers a whole new dimension of space to work with. As with all progressions of technology, this added a dimension of interactivity; therefore, this requires UI guidelines to be reiterated [13].

Conventionally, 2D UI components such as menus, windows, and scroll bars are widely used for clicking, zooming, and dragging, and they are proven to be proficient at these tasks. On the other hand, 3D UIs, such as virtual hands, in-air gestures, and gaze, often offer six degrees of freedom (DoF) or more in its interface to facilitate common user tasks such as: point-to-select, grasp-to-manipulate, navigation, and system control. In terms of devices, various hardware such as mouse/keyboard, joystick, touch surfaces, and an array of sensors including gesture and gaze sensors, facilitate user–computer interaction. Each of these devices is best suited for specific scenarios and comes with its own set of limitations.

Unfortunately, designing user-friendly UIs for VR applications is challenging due to there being numerous factors that impact user experience (UX). The wide variation in input devices, task requirements, and display constraints often leads to mismatches, making it difficult to create an optimal UI in the VR environment. Many existing VR application user interfaces consist of 2D UI components directly mapped onto a 3D environment. This approach might not be ideal for UX as humans consume information differently in 2D and 3D interfaces, making it unlikely that best practices for 2D UI design will always translate or apply in a 3D context. For example, it is not always easy to select a 2D menu using an in-air controller.

Furthermore, certain functions, like gestures with multiple DoFs, are often underutilized and are commonly relegated to simple icon pointing in many indirect interactions. However, exploiting these functions to their full potential can sometimes result in excessive freedom, leading to frustration, distraction, or inaccuracies. For example, navigating a 2D map can be significantly more straightforward than 3D teleportation in an X-Ray style half transparent 3D environment.

Numerous articles discuss the best approach for immersive product design, 2D or 3D first designs [14], and categorizing the VR UIs based on many factors like task decomposition [15] and metaphors [16]. After reviewing the eligible papers, two main overarching types of 3D UI in VR were observed. These are the implementation of *2D UIs in a 3D environment* and *3D UIs based on reality-based interactions*.

Different to the existing approach, a simpler and more fundamental model is proposed to enhance future proposed 3D UI design guidelines. This paper aims to consolidate efforts in 3D UI development, facilitating the easy description, sharing, and implementation of 3D UI design. Its main idea is inspired by the process of developing a 3D environment or interaction. In this process, the very first step is always establishing the axis system. This dimension serves as the basic perspective upon which the entire VR world and its interactions are constructed. With this spirit, this survey focuses on discussing the 3D UI work from a dimensional perspective. Its simplicity and fundamental nature would provide

an effective point of view from which to study 3D UIs. This could potentially benefit the refinement and reiteration of design guidelines for 3D UIs. The main contributions are thus as follows:

1.  A novel metaphor-guided quadrant model to delineate the 3D UI problem in an end-to-end manner from the dimensional perspective is proposed.
2.  Thirty-three unique articles selected using a meta-analyses methodology were reviewed.
3.  Reiterate design recommendations based on the proposed model.

## 2. Approach

### 2.1. Proposed Categorization Model

Existing categorization approaches are based on task decomposition [15], metaphors [16], a set of relationships including selection tool-control, control–display ratio and motor–visual space relationship [17], input-control feedback [18], and spatial transformation decomposition [19]. Other surveys are similar in spirit and provide comprehensive considerations and perspectives for VR and virtual environment elements [7,14,20–24].

Different from these approaches, an end-to-end metaphor-guided quadrant model is proposed (Table 1), which directly addresses user input and targets virtual environments, focusing on their intrinsic dimensional characteristics. This simplicity and the direct approach make it a practical framework for analyzing commonly used 3D UIs, facilitating the selection of the most appropriate technique for a specific task.

**Input dimensional semantics:** Conceptually, this refers to the dimension of the input from the user's intentional point of view, that is, the user can achieve 2D or 3D effects or results directly in the UI with the input methods. This follows the end-to-end theme, with the focus on the end effects of the input.

For example, the typical 2D user input methods are button, keyboard/mouse, and touch surface. For input dimensional semantics, these input methods are essentially planar in nature with 1 or 2 DoFs. Semantic-wise, they are toggle on/off, increase/decrease, pressing, typing, and 2D plane pointing/transformation. The typical 3D user input methods are the in-air controller and gaze. For input dimensional semantics, these inputs are essentially 3D in nature with 3 to 6 DoFs; semantic-wise, they are 3D pointing and transformation.

However, the input methods, such as the joystick (with more multi-axis (3D) and buttons (2D)) and gesture (with (3D) or without depth (2D)), are seen as both 2D and 3D; these input methods can achieve both 2D and 3D effects in nature.

**Target outcomes:** The outcomes of virtual targets are also categorized into 2D and 3D, that is, the results of the interaction would cause the states of the virtual targets to change in 2 (1 or 2 DoFs) or 3 (3 to 6 DoFs) dimensions or generate 2D or 3D data. The interactions, according to their metaphors as shown in Table 2, are mapped into this input-outcome quadrant model as shown in Table 1, while the interaction's input semantics as previously described is also incorporated. For example, for the menu, the target button or option has 2D outcomes and its stages are selected/unselected, and for the grasp, the virtual objects would be transformed in 3D space. Note that in Table 1, if the interaction could generate eight 2D (or less) or 3D (or more) outcomes, it spans across both columns; similarly, if it has the 2D or 3D input semantics then it spans the two rows.

**Table 1.** The proposed metaphor-guided end-to-end dimensional quadrant categorization model.

| Input Semantics / Virtual Target Outcomes | 2D or Less | 3D or More |
|---|---|---|
| 2-Dimensional Input | Menu, Typing | Indirect |
| | Surface | Point |
| 3-Dimensional Input | | Navigation |
| | | Grasp |

**Table 2.** Common interaction metaphors.

| | |
|---|---|
| Menu | Select or logically browse through a list of options or commands. |
| Typing | Add, edit or delete texts for textural interaction. |
| Point | Point at on-screen (2D) or virtual (3D) elements through ray or virtual hand. Discussed in 3D. |
| Surface | Manipulate virtual objects in a 2D context like (axis-aligned) dragging or rotating (additional touches for 3D). Discussed in 2D |
| Grasp | Manipulate virtual objects in a 3D context. |
| Indirect | Manipulate virtual objects indirectly through handles and mappings. |
| Navigation | Control the virtual cameras. |

### 2.2. Survey Approach

Papers published that designed, surveyed, or reviewed 3D UIs for VR were sourced using the search engine Google Scholar, using the keywords "3D", "user interface", "design", and "guidelines", and their corresponding abbreviations and synonyms. Other criteria included more than ten citations for publications older than ten years, and contained related reported results on the design of 3D UIs. Articles that address the design of 2D/3D UIs in/for 3D environments, help with answering the research questions or reiterating design guidelines and, as this work is a conjuncture of HCI and VR, highly cited works from reputable journals or conferences such as ACM's CHI [25–32], IEEE's VR and TVCG [21,22,33–44], as well as other conference and journal sources [13,45–58] from between the early 2000s to 2023 were selected.

This meta-review will aim to provide an updated overview of significant works in the field that relate to 3D UIs in VR to establish an end-to-end dimensional perspective for a set of reiterated 3D UI design recommendations to guide future development.

### 2.3. Research Questions

The use of 2D UI in a 3D space raises the challenge of determining which design(s), whether 2D or 3D, will optimize the UX in VR. This raises two questions. First, how different are the interaction modalities for determining the design of 2D and 3D UIs for VR? And what are the designs that have been tested but have yet to validate the claim of an optimized UX in 3D VR? The first question examines the common forms of input and interactions with virtual targets when in a 3D environment. The second question seeks to determine a set of recommendations for 3D UI designs through a publication review.

## 3. Input Dimensional Semantics

This section discusses the dimension of the input from the user's point of view, with the user achieving 2D or 3D results with the input methods.

### 3.1. 2D Input

Design principles established for 2D UI based on WIMP standards [59] are still an efficient and common method of designing 3D UIs [54]. These implementations generally take the form of a floating widget or window that the user can interact with using a pointer. The pointer can be a ray cast from the HMD's head tracking or controller.

There are various interpretations of 2D UI in VR environments; however, the most common is to arrange the UI displayed as a plane, similar to how it would be displayed on a flat display. This display is then represented in the 3D environment as a floating widget. The significance of this method is that it is familiar to most users who have been exposed to modern technology, such as a computer or smartphone; the most commonly used functions are displayed directly before the user. This, while neater, however still makes use of the traditional 2D medium, and the 3D space is not utilized. Moreover, fewer icons/functions

can be displayed as there is a limit before the UI becomes too cluttered for users to have ease of use.

Another common method is to overlay the UI on the HMD's field of view, known as the head up display (HUD). This method will require users to perform the physical action of moving their heads in search of desired functions, though it utilizes the 3D space which enlists its eligibility as a 3D UI, and provides immersion to the users. Despite this, it has its disadvantages as well, mainly the need to find the balance between the size of the UI and also the placement of the UI by the user to avoid the UI clipping into the environment if users are moving.

Controller or pointer-based input is used to navigate menus on these displays as the user would on a computer or console. These 2D design principles have been used and validated for decades, being able to leverage metrics like Fitts' Law [60] to calculate the performance and efficiency of the selection method.

*3.2. 3D Input*

Human–computer interaction (HCI) occurring in 3D space does not necessarily define the term *3D interaction* as it may not constitute a 3D input [61,62]; such interaction may involve only 2D input, in which users accomplish tasks using a traditional 2D UI. Conversely, using hardware to translate a 3D object **can** be defined as a 3D interaction. Thereby, as defined by Jacob et al. [28], 3D UI implementations involve interacting with objects in the virtual environment, similar to how humans would interact with objects in the real world. These implementations use humans' knowledge of physics and their bodies, and their social and environmental awareness when interacting with the world around them. These real-world heuristic affordances are, on some level, intuitively transferable to 3D UIs [55]. Generally, these implementations are deemed to take more effort to execute compared to 2D UI as they require more complex gestures and input from the user. However, multiple studies have found that RBI 3D UIs increases user engagement and is seen as fun. However, a counter-argument could be made that the hassle of the UI might hinder the user's efficiency when performing tasks with RBI.

**4. Outcomes of Virtual Targets**

The outcomes of the input interactions based on their metaphors are discussed here. The results of the interaction would cause the states of the virtual targets to change dimensions or create 2D or 3D data.

*4.1. 2D Interfaces*

When placing a 2D interface in a 3D environment, one factor which determines how readable the UI will be is the distance from the camera and how large the font of the UI is. The efficacy and comfort of 3D visualizations are intricately linked to the depth perception of 3D human visual perception [53].

Steinicke et al. [27] found that VR using stereoscopic projections provided additional depth perception for users, but the challenges and limitations of touch interaction had not been sufficiently investigated. Zhou et al. [45]'s recommendation for ergonomic arrangement for a 3D UI in a virtual environment was to place it 3.5 m away from the user to have the best viewing experience. For a 3D UI that the user can interact with, a distance of 2.6 m was recommended. They also found the optimum depression angle for operation UIs is around 9.8 degrees, fitting a user's comfortable field-of-view and hand-controlling posture.

4.1.1. Menus

Menus are visual or interactive lists of options that users can choose from to navigate through or interact with a system. In a 2D UI, there are various forms of menus such as dropdown menus, navigation menus, hamburger menus, toolbar menus, pie/ring menus [63], and context menus. When these menus are translated to a 3D UI in VR, these elements consistently preserve their functionality despite the shift to a 3D display paradigm.

Teylingen [64] proposed a *virtual data visualizer* (VDV), which is representative of floating menus, a common method for displaying menus across different platforms even today. Bowman and Wingrave [65] designed the *three-up, labels in palm (TULIP)* menu system, which utilizes the palm to display the menu. Mine [66] explored the worlds-in-miniature (WIM), in which a user can directly interact with 3D virtual objects scaled down to users' peripherals. Developed by Liang and Green [63], geometric objects are arranged in the 1-DOF ring formation where a flick of the wrist will rotate the selection. In Gerber and Bechmann [67]'s design, objects are arranged in a circle paradigm and the wrist is rotated in the horizontal plane to make the selection. The look-at menu by Mine et al. [68] uses head movement instead of hands to interact with the menu.

Dachselt and Hübner [54] surveyed multiple 3D menu solutions (see Figure 1) and defined a taxonomy for 3D menus (temporary option menus, single menus, menu systems, and menu hierarchies) in their survey of 3D menus, as seen in Table 3.

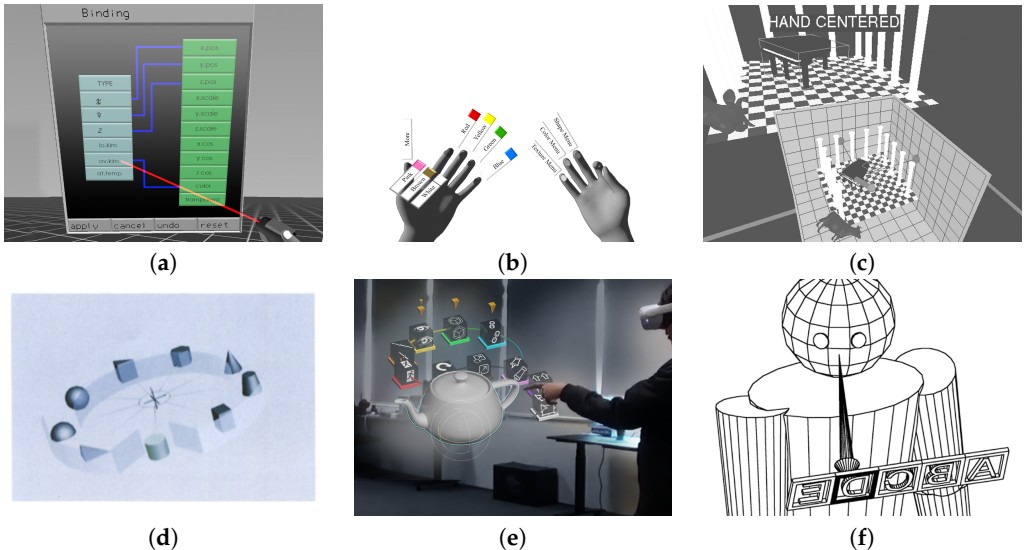

|  (a) | (b) | (c) |
| (d) | (e) | (f) |

**Figure 1.** Examples of VR 3D Menus from left to right: (**a**) floating menu [64], (**b**) TULIP menu [65], (**c**) worlds-in-miniature [66], (**d**) ring menu [63], (**e**) spin menu [67], (**f**) look-at-menu [68].

**Table 3.** Dachselt and Hübner's taxonomy of 3D menus according to the criteria's hierarchical nature and structural layout.

| Hierarchical Nature | Structural Layout |
| --- | --- |
| Temporary Option Menus | List, Rings, Matrix, Geometric structure |
| Single Menus | List, Rings, Matrix, Geometric structure, Free layout |
| Menu Systems | List, Rings, Geometric structure |
| Menu Hierarchies | List, Rings, Geometric structure, Free layout |

They identified five examples of structural layouts: lists (Figure 1a); rings (Figure 1d,e); matrices (Figure 1b); geometric structures (Figure 1c); and free layouts (Figure 1f), which correlate to different hierarchies. Their taxonomy prepared the ground for further classification, and could be applied to the evaluation of menu suitability for VR applications.

4.1.2. Typing

Typing is an essential input method used in 2D UIs in scenarios that require a complex chain of inputs (Figure 2). This is generally reserved for the input of language and communication. The metric in which researchers calculate the efficiency of typing is words per minute (WPM), with an average of 60–70 WPM with traditional keyboard typing [69]. With VR, the WPM across different techniques (raycasting, drum-like VR keyboard, head-directed and split keyboard) was significantly slower at 10–20 WPM [70].

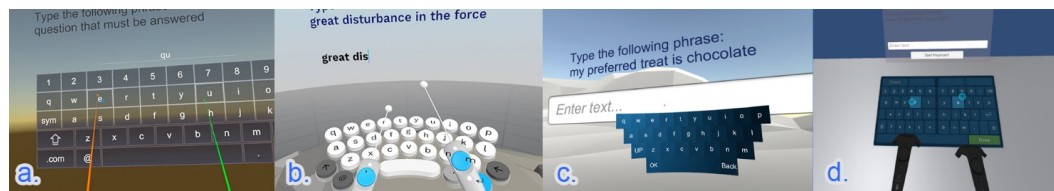

**Figure 2.** Example of controller typing from left to right: (**a**) raycasting; (**b**) drum-like keyboard; (**c**) head-directed input; and (**d**) split keyboard by Boletsis and Kongsvik [70].

Compared to traditional typing on a QWERTY keyboard, where the user can leverage touch typing with multiple fingers and muscle memory, typing in VR is usually limited to one-to-two input controllers pointed at floating widgets.

### 4.1.3. Surface

With the rapid development of multi-touch techniques as used in phones and tablets, surface- or touch-based interaction has become very popular. It provides direct interaction and it is typically used to drag and rotate virtual objects in 2D contexts. However, additional touches can be introduced to drag or rotate virtual objects in 3D contexts; this may not always be intuitive and convenient, as it is essentially based on 2D surfaces [71–73]. As shown in Figure 3a–i, such surface-based interaction can be applied in many applications (refer to the survey [74]), such as architectural modelling [75], data exploration [76], architectural walkthrough [77], and sand drawing [78].

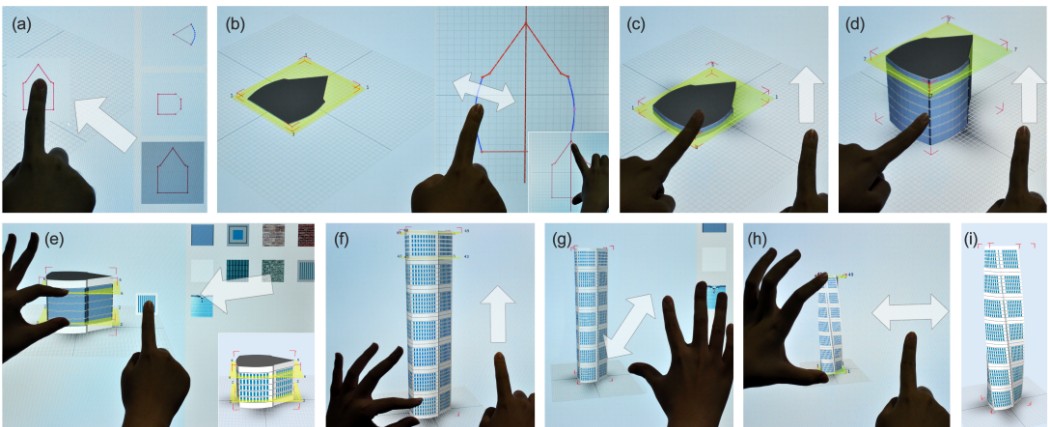

**Figure 3.** Surface-based 3D manipulation for architectural modelling by Sun et al. [75].

Similarly, surface UIs in 3D use 2D planes to display information similar to what would be displayed on a monitor or screen. These are either represented as floating widgets or are incorporated in the virtual environment. When integrated into the 3D environment, there has been some research into adding physical props to aid with the tactile feedback of interacting with a surface, carried out by Lindeman et al. [41], as seen in Figure 4. However, most designs are not implemented literally, as their study opted for floating widgets or planes, even with the lack of physical feedback.

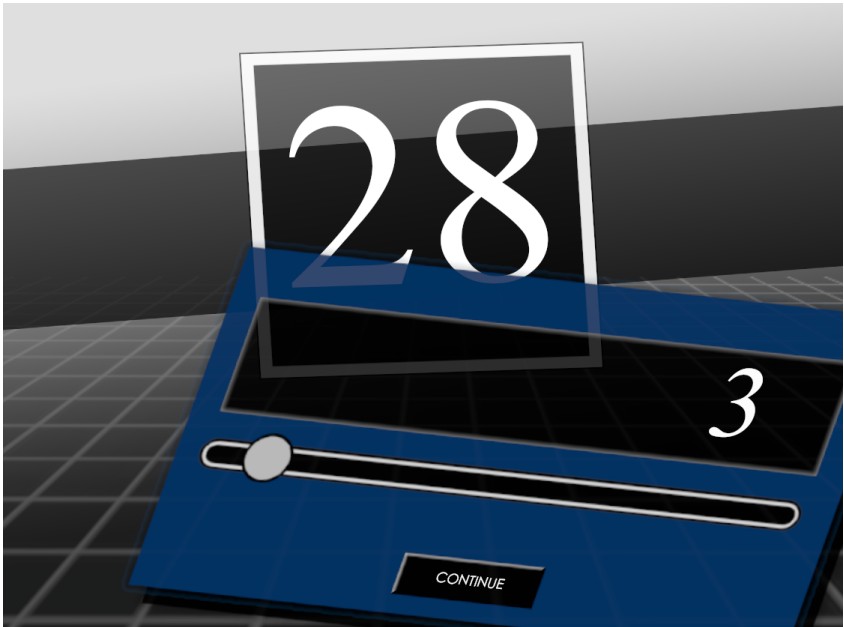

**Figure 4.** Example of Lindeman et al. [41] surface 3D UI in simulation.

### 4.2. 3D Interfaces

In VR simulations, the main forms of 3D interactions are manipulation, navigation, communication, and system control [79,80]. These interactions comprise all the ways the user will interact with the VR simulations, from interacting with the environment to the system and other entities or users in VR.

3D UI selection techniques should be accurate, easy to understand, rapid, and not take too much energy [17]. Jacob et al. [28] defined a framework, reality-based interaction (RBI), which uses our pre-existing understanding of the world to interact with elements in a virtual environment.

Weiß et al. [81] and Hepperle et al. [82] studied which type of UI is most advantageous when tasked with interacting with different tasks. They found that 2D UIs performed the best in tasks involving manipulating multiple objects rapidly and accurately while 3D UIs were associated with fun and immersiveness.

Previous surveys on 3D UI interaction techniques were conducted from 2012–2013 [17,21]. At the time, VR technology was not as prevalent as today; however, technology like the Kinect and WiiMote could also utilise 3D UI.

#### 4.2.1. Point

Object interaction in a 3D environment requires the indication of the desired target, assuming said object had been programmed to be identifiable by the system, and pointing accomplishes that efficiently. The typical input methods come in the form of mouse, trackball, motion gestures, and controllers [19,43,66]. A virtual hand is typically used to select the object in VR that is within reach [83]. If object selection falls beyond arms' reach, a popular method in VR is the use of raycasting [31] from the controller or the HMD displayed, where a beam is projected as a guide for users, to more accurately control and visualise the selection process; it is also used for pointing 2D menus.

In a study by Dachselt and Hübner [54], they surveyed 3D pointing techniques in VR and found that Fitts' Law [60] was used frequently as the measure for pointing tasks and implementations. Fitts' Law was an equation used to predict the time required for users to move their pointer from the start to the target position as a function of the distance and size of the target. Grossman and Balakrishnan [84] later reviewed and adapted the original

equation, meant to measure 2D space in 3D environments. The 3D Fitts' Law equation by Grossman and Balakrishnan [84] (the parameters are adopted from their study):

$$MT \approx 56 + 508 \cdot \log_2\left(\sqrt{f_W(\theta)\left(\frac{A}{W}\right)^2 + \frac{1}{9.2}\left(\frac{A}{H}\right)^2 + f_D(\theta)\left(\frac{A}{D}\right)^2} + 1\right), \quad (1)$$

where $f_W(0°) = 0.211$, $f_W(90°) = 0.717$, $f_W(45°) = 0.242$, $f_D(0°) = 0.194$, $f_D(90°) = 0.312$, $f_D(45°) = 0.147$. $A$ is the distance to target and $W$, $H$, and $D$ are the width, height, and depth, respectively.

One issue with raycasting was that the source of the ray from either the controller or the HMD could produce problems like solid-angle mismatch, occlusion mismatch at object level, and occlusion mismatch at point level, as shown in Figure 5.

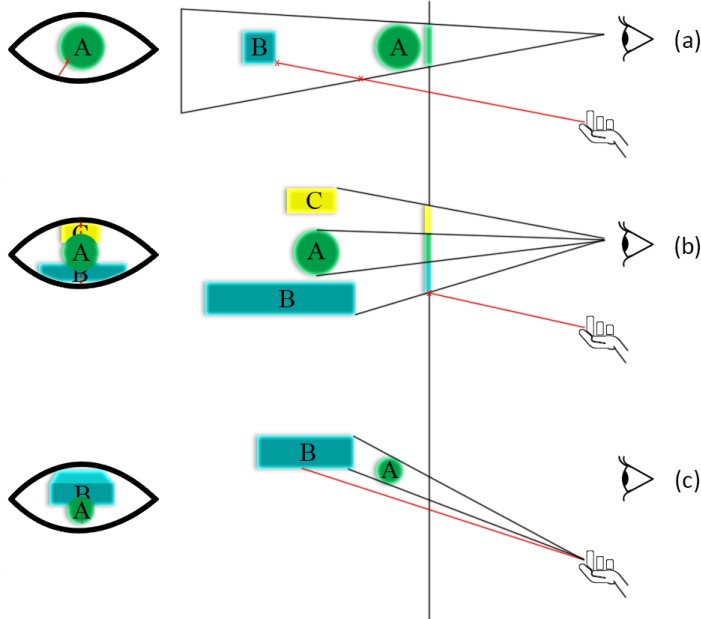

**Figure 5.** Example of (**a**) solid-angle mismatch; (**b**) occlusion mismatch at object level; and (**c**) occlusion mismatch at point level from Argelaguet and Andujar [42].

Argelaguet and Andujar [42] performed a study comparing different pointing selection methods (hand-rooted and eye-rooted selection techniques) in VR and found that a hybrid approach using HOMER (hand-centered object manipulation extending raycasting) [32] was the most efficient.

### 4.2.2. Grasp

Grasping refers to the interaction between a user and virtual objects within the environment. It involves the ability to reach out and grab, hold, or manipulate virtual objects. This interaction is typically facilitated by motion controllers, gloves, or other hand-tracking devices that allow users to manipulate virtual objects as if they were interacting with objects in the real world.

Humans have an innate understanding of how to grasp and manipulate objects in the real world. However, in VR, these actions are often predefined and limited, and executed through animations due to the complex and multitude of ways a user can decide to interact with an object in their hands. Oprea et al. [56] proposed a method to grasp and manipulate an object regardless of its geometry in an effort to address this limitation using handheld controllers. A later study by Shi et al. [44] proposed another method using a three-module approach to grasping (grasping detection, hand–object connection module, and finger bending) that addresses some of the deficiencies discovered. Their

implementation improved the realistic rendering of the 3D hand model when interacting with objects in VR, as seen in Figure 6.

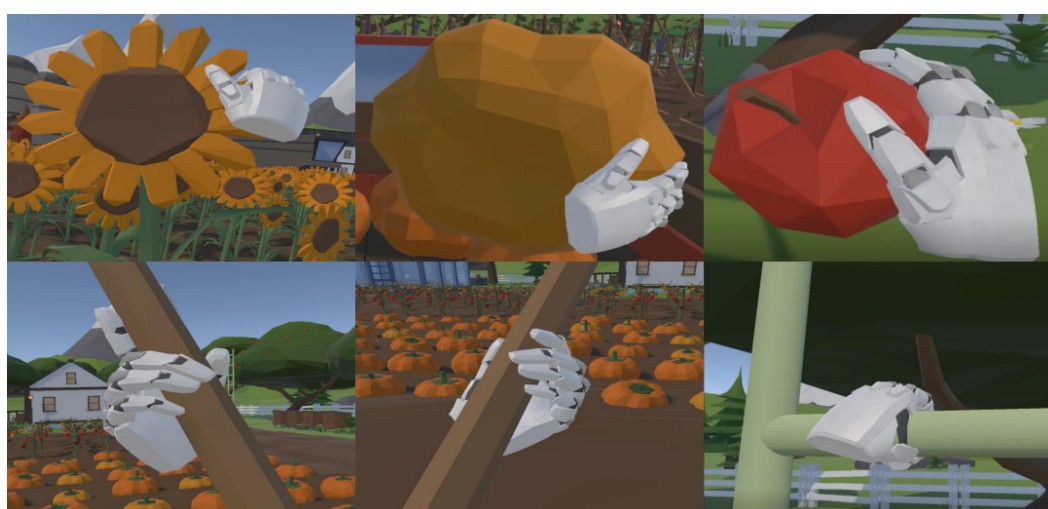

**Figure 6.** Example of grasp interactions with VR objects by Shi et al. [44].

### 4.2.3. Indirect

These refer to indirect mappings between the controls and elements in the user interface and the target actions or outcomes they represent. Good spatial mappings are desirable to create a clear connection between the input and the system's output. This type of interaction is typically based on additional handles or cues, and the common input methods for VR are in-air controllers and gestures; however, some additional handlers may not be easy and natural to use and map, and in-air gestures often suffer from accuracy drawbacks [19,74]. As shown in Figure 7, additional metaphors are usually introduced to facilitate and map the manipulation, such as bars [85], pins [86,87], virtual handles [88], and widgets [89].

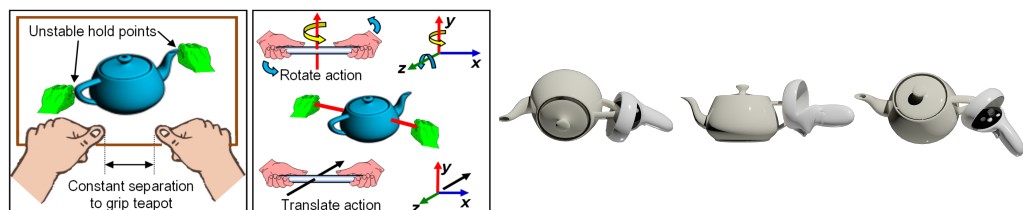

**Figure 7.** Example of indirect 3D manipulation based on bar [85] and pin [87] handlers.

### 4.2.4. Navigation

Navigation is a form of exploration or wayfinding in order to perform given tasks [90]. Depending on the UI chosen, 2D WIMP-based implementations follow a conventional style of input using buttons and/or pointers, while 3D RBI-based implementations have additional virtual spatial requirements such as physically reaching out to interact with the object in virtual space. The conventional methods used in many VR games and simulators are based on steering (e.g., first-person shooting game) and teleporting using controllers or joysticks or gaze [91,92] (Figure 8). Additional hardware may be utilized for walking, cycling, driving, and flying in a 3D environment [29,93] (Figure 8), and 3D navigation may not always be easy, usually requiring additional cues [94,95], compared to 2D map-based navigation, which provides a better overview [96].

During task execution, efficiency and accuracy are important while navigating through 3D UIs. Similar to pointing, the metric most often associated with navigation efficiency is Fitts' Law. Navigation is also intrinsically linked to the type of menu employed by the simulation.

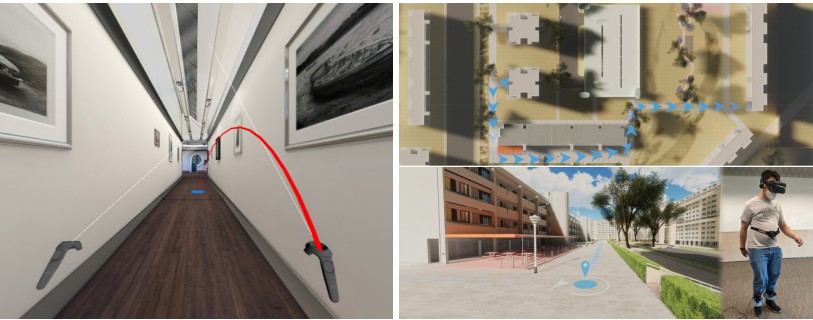

**Figure 8.** Navigation by teleporting [91] and walking [29].

## 5. Discussion

### 5.1. Applications

An important determining factor in choosing which 3D UI would best serve the application is the context in which the user interacts with it such as education, simulation, and entertainment. While education and simulation applications tend to be more serious, informative, and accurate, applications that are designed for entertainment can take creative liberties in their portrayal of the design [48].

The idea of immersion in the education and simulation setting is intrinsically tied to the dynamic interaction between users and the VR environment. The user interface serves as the pivotal point for user interaction within the VR teaching context, which influences the overall quality of teaching and learning. The design of the user interface holds considerable sway over learners' ability to grasp target knowledge efficiently. Additionally, it directly affects learners' motivation, concentration, and active participation in the learning process. While educational VR strives to be accurate and informative, this does not necessarily mean it cannot be fun. Gamification in educational VR has proven to be as effective at learning outcomes as traditional learning methods. Bedwell et al. [97] proposed a framework for gamification elements that could be implemented in non-game activities.

### 5.2. Reiterated Design Recommendations

In the exploration of 3D UIs, a key insight emerged: the consideration of dimensional affordance heuristics was crucial in the design of 3D UIs. These heuristics encompassed factors such as perspective errors and the absence of physical feedback. In contrast to traditional 2D user interfaces, which are confined to a flat plane, 3D UIs introduced heightened complexity in terms of both implementation and user experience. This review distilled and suggested reiterating three primary recommendations, matching devices to interaction and to the tasks in 3D environments, reducing degrees of freedom (DoFs) and imposing constraints when possible, and:

(1) Matching devices to interaction and to the tasks in 3D environments. The recommendation is to align the dimensionality of input semantics with that of the virtual target outcomes, as matching these dimensions can result in an intuitive and natural user interface. For instance, in-air controllers (3D input semantics) are highly effective for tasks like 3D pointing and grasping (3D target outcomes). However, for 2D tasks, such as 2D pointing and typing (2D target outcomes), in-air controllers may be underutilized or less effective. In such cases, designing appropriate layouts like in-position widgets, opting for button-based interfaces, adding constraints, or providing cues could be more suitable alternatives;

(2) Reducing degrees-of-freedom (DoFs) and imposing constraints when possible. To avoid the frustration and distraction caused by excessive freedoms, and considering the frequent mismatches between input semantics and target outcomes, it is advisable to impose constraints to reduce the DoFs. For instance, when using in-air controllers (3D input semantics) for a surface task such as 2D dragging (2D target outcomes), limiting the motions in the axis aligned to reduce the input semantics to 2D would make the interaction more effective;

(3) No single best interaction technique. The suitability of interaction techniques in VR varies depending on the types of tasks, particularly from a dimensional perspective. While no single solution fits all scenarios, it is recommended to assess the dimensional nature of each task and, accordingly, select the most appropriate interaction technique or a hybrid approach. For instance, using an in-air controller for 3D selection, but voice input for typing.

*5.3. Challenges and Future Direction*

Quick prototyping for 3D UIs is not readily available, often necessitating direct integration into VR engines. Although Fitts' law, initially established for 2D UI efficiency, remains relevant for 3D UIs, the design of 3D UIs largely depends on the specific application's purpose, style, fidelity, and interactivity. One future direction would be consolidating standardized and comprehensive 3D UI design guidelines. Thus, the proposed approach is the first step towards this goal. Although there is some existing work [98,99], additional empirical studies comparing the effectiveness across 3D UIs based on the abovementioned guidelines could be validated.

VR environments, merging 2D imagery with real-world display techniques, challenge human visual physiology, leading to user discomfort issues such as dizziness, visual fatigue, and more severe symptoms like eye swelling and nausea. Unlike traditional screen interfaces that focus on functionality, VR UI design should prioritize interactive experience; however, mitigating such comfortableness would increase user satisfaction level; one possible solution based on the proposed model would be choosing a suitable dimension in inputs and outcomes to reduce unnecessary distraction and fatigue.

A notable limitation of VR UIs is the lack of adequate feedback. Traditional UIs use visual and audio cues, and advancements have introduced haptic feedback in gaming controllers. However, current haptic feedback, mainly vibrations, provides limited physical sensation. This can lead to a disconnection in 3D environments, where users might expect physical resistance that haptic feedback cannot fully replicate. Pseudo-haptic solutions, leveraging visuohaptic illusions that utilize visual dominance over haptic perception, could be an effective alternative or complement [36].

Looking ahead, VR UI design should explore multi-modal or hybrid methods, incorporating human communication techniques, such as speech and brain signals, to enhance user interaction and immersion. Based on the proposed model, this can increase dimensions in VR UIs, offering greater design flexibility of 3D UIs to enhance user experience.

## 6. Conclusions

In this review, an overarching view of the research into 3D UIs was given and common themes and recommendations from different areas of focus were distilled. A metaphor-guided quadrant model based on the input semantics and the target outcomes in 2D and 3D was proposed, as seen in Table 1, as a means to categorize their implementation alignment. It directly addresses user input and targets virtual environments with an emphasis on their intrinsic dimensional characteristics. This end-to-end and simple approach provides a practical framework for analyzing commonly used 3D UIs. Three design recommendations were proposed based on the considerations of dimensional affordance heuristics uncovered for researchers to consider when designing future 3D UIs for VR applications based on the insights from the review.

**Author Contributions:** Conceptualization, A.Y., A.J. and J.S.A.L.; methodology, A.Y., B.W.J.K., A.J., K.C. and J.S.A.L.; validation, A.Y., B.W.J.K., K.C. and J.S.A.L.; formal analysis, A.Y., B.W.J.K., K.C. and J.S.A.L.; investigation, A.Y. and A.J.; resources, A.Y.; data curation, J.S.A.L.; writing—original draft preparation, A.Y. and B.W.J.K.; writing—review and editing, A.Y., B.W.J.K., K.C and J.S.A.L.; visualization, B.W.J.K.; supervision, J.S.A.L.; project administration, J.S.A.L. All authors have read and agreed to the published version of the manuscript.

**Funding:** Part of this research was supported by the National Research Foundation (NRF), Singapore, and the Singapore Land Transport Authority (LTA), Singapore, under the Urban Mobility Grand Challenge (UMGC-L010) Grant.

**Data Availability Statement:** Data sharing not applicable.

**Conflicts of Interest:** The authors declare no conflict of interest. Any opinions, findings and conclusions or recommendations expressed in this material are those of the author(s) and do not reflect the views of National Research Foundation, Singapore and the Singapore Land Transport Authority.

## Abbreviations

The following abbreviations are used in this manuscript:

| | |
|---|---|
| VR | Virtual Reality |
| HMD | Head Mounted Display |
| WIMP | Windows, Icons, Menus, Pointers |
| 3D | 3 Dimensional |
| 2D | 2 Dimensional |
| UI | User Interface |
| DoF | Degrees of Freedom |
| UX | User Experience |
| HUD | Head's Up Display |
| HCI | Human Computer Interaction |
| VDV | Virtual Data Visualizer |
| TULIP | Three Up Labels In Palm |
| WIM | Worlds in Miniature |
| WPM | Words Per Minute |
| RBI | Reality Based Interaction |
| HOMER | Hand-Centered Object Manipulation Extending Ray Casting |

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
