# Peer review of "Entering the Next Dimension: A Review of 3D User Interfaces for Virtual Reality"

_electronics, doi:10.3390/electronics13030600_

Round 1

Reviewer 1 Report

Comments and Suggestions for Authors

The manuscript is detailed and provides a decent overview of the current sate of 3D user interfaces for virtual reality. I especially believe it is a great primer for anyone who is not an expert in the use of such environments - as the manuscript is very approachable. Still, there is a number of comments that need to be addressed, in order for the manuscript to be clearer and more understandable.

1. Table 2 is placed near the start of the manuscript, but only referenced in the Conclusion. In addition to that, it is poorly explained, and as such needs to be addressed in greater detail.

2. Table 3 is unnecessary as the table, and can be given in textual form. 

3. All tables should have their titles placed above the table.

4. L.124 "...and were filtered through a selection criteria process." - Due to this being a review paper, the details of the the criteria used for paper selection need to be given in greater detail. Were they just the ones listed in the same paragraph below or are there more? How do the researchers qualify "highly cited papers" - is there a citation cut-off and how is it determined?

5. Figure 1 is hard to see in the given form, and the subfigures should be larger - maybe place them in two rows?

6. Table 4. Structural layouts given as "list, rings. matrix, geometric structure" is unclear to people who aren't involved in VR use and/or production. Authors should at least briefly describe or show examples of these. 

7. Equation after L. 276 us not numbered.

8. Related to the above equations - values for fw given below - how are they obtained? Are they calculated with an equation - in which case it should be added, or just experimental? Please clarify. Also, I believe it would be more readable to include these values in the table.

9. There is a number of spacing errors, where the space needs to be added after the end of sentence, e.g. L. 277

Due to the above comments, I suggest the publication of the manuscript after minor revisions are performed by the authors.

Reviewer 2 Report

Comments and Suggestions for Authors

1.      This paper presents a research study conducted through a literature review and analysis. Initially, a conceptual framework for analysis is proposed, followed by an examination of input and output modalities in the VR environment. Finally, principles and recommendations for UI design application within the VR environment are put forward.

2.      The research introduces specific research questions and criticizes the limitations of applying 2D UI in a 3D environment, thereby highlighting the primary contributions of this study. Figures are also incorporated throughout the paper to facilitate readers' understanding.

3.      The research analyzes various interaction modes derived from over 30 relevant literature. However, it remains unclear from the author's exposition whether these interface modes have been empirically confirmed in terms of the efficiency and effectiveness in related studies. Please explain appropriately if possible.

4.      The study consolidates various 3D UI applications, yet potential limitations may persist. In section 5.2, the authors propose design recommendations. Could these can be compared with some existing empirical studies and identified? Or could they be validated in future research endeavors?

5.      Overall, this appears to be a significant research paper in the domain of 3D UI, offering valuable contributions to system development in related fields. I recommend accept with minor revisions.

Reviewer 3 Report

Comments and Suggestions for Authors

The manuscript is interesting and easy to follow. However, we think that the authors should cover some issues that have not been properly addressed, according to our viewpoint.

  • First of all, authors should keep in mind that when preparing a manuscript, it is common practice to include an "Abbreviations" section at the end of the document, right before the references. It is important to note that in the abstract section, abbreviations are typically avoided, and their first introduction occurs in the introduction section or later in the manuscript.

  • The authors need to expand the introduction section to clarify the need for their study. What is missing and how this study can fill in that gap should be described.

  • We think that a related work section should be included in order to better motivate the direction of the new research. Considering this set of previously developed works, what is the motivation for the presented research? The specific contribution to the state of the art should be highlighted.
  • Please enrich the conclusions section in order to reveal the final outcome of the manuscript.
  • Finally, consider adding some publications to the related work section. For example:
  • Trichopoulos, George, et al. "ARTISTS: A virtual Reality culTural experIence perSonalized arTworks System: The “Children Concert” painting case study." Proceedings of the International Conference on Digital Culture & AudioVisual Challenges (DCAC-2018). 2018.
  • Yue, Gu. "3D User Interface in Virtual Reality." International Conference on Human-Computer Interaction. Cham: Springer International Publishing, 2021.
